# A *Salmonella* Microfluidic Chip Combining Non-Contact Eddy Heater and 3D Fan-Shaped Mixer with Recombinase Aided Amplification

**DOI:** 10.3390/bios12090726

**Published:** 2022-09-05

**Authors:** Shangyi Wu, Hong Duan, Yingchao Zhang, Siyuan Wang, Lingyan Zheng, Gaozhe Cai, Jianhan Lin, Xiqing Yue

**Affiliations:** 1College of Food Science, Shenyang Agricultural University, Shenyang 110866, China; 2Key Laboratory of Agricultural Information Acquisition Technology, Ministry of Agriculture and Rural Affairs, China Agricultural University, Beijing 100083, China; 3Beijing Engineering and Technology Research Center of Food Additives, Beijing Technology & Business University, Beijing 100048, China; 4State Key Laboratory of Transducer Technology, Shanghai Institute of Microsystem and Information Technology, Chinese Academy of Sciences, Shanghai 200050, China

**Keywords:** microfluidic chip, eddy heating, 3D fan-shaped mixer, recombinase-aided amplification, *Salmonella* detection

## Abstract

Foodborne pathogenic bacteria have become a worldwide threat to human health, and rapid and sensitive bacterial detection methods are urgently needed. In this study, a facile microfluidic chip was developed and combined with recombinase-aided amplification (RAA) for rapid and sensitive detection of *Salmonella typhimurium* using a non-contact eddy heater for dynamic lysis of bacterial cells and a 3D-printed fan-shaped active mixer for continuous-flow mixing. First, the bacterial sample was injected into the chip to flow through the spiral channel coiling around an iron rod under an alternating electromagnetic field, resulting in the dynamic lysis of bacterial cells by this non-contact eddy heater to release their nucleic acids. After cooling to ~75 °C, these nucleic acids were continuous-flow mixed with magnetic silica beads using the fan-shaped mixer and captured in the separation chamber using a magnet. Finally, the captured nucleic acids were eluted by the eluent from the beads to flow into the detection chamber, followed by RAA detection of nucleic acids to determine the bacterial amount. Under the optimal conditions, this microfluidic chip was able to quantitatively detect *Salmonella typhimurium* from 1.1 × 10^2^ to 1.1 × 10^5^ CFU/mL in 40 min with a detection limit of 89 CFU/mL and might be prospective to offer a simple, low-cost, fast and specific bacterial detection technique for ensuring food safety.

## 1. Introduction

The outbreak and prevalence of foodborne diseases not only place a substantial burden on global healthcare systems but also have a serious impact on economic and social stability [1]. Globally, one in ten people is affected by foodborne pathogens annually. Among these pathogens, *Salmonella* is the leading cause [2], and typhoid fever is a life-threatening infection caused by *Salmonella typhimurium* [3]. Since food contamination may occur at every stage of food supply chains from field to table, monitoring and identification of contaminated foods are crucial to safeguard food supply chains [4,5]. To date, the gold-standard culture method is still the most reliable way to detect bacteria, but it is time-consuming and not suitable for on-site monitoring [6,7]. Other recommended methods, such as enzyme-linked immunosorbent assay (ELISA) and polymerase chain reaction (PCR), are time-saving, but ELISA has to screen specific antibodies and PCR requires expensive equipment [8,9,10]. Therefore, fast, accurate, low-cost and in-field detection techniques are in urgent need.

In recent years, various isothermal nucleic acid amplification technologies have received widespread attention because they exhibit molecule detection features such as accuracy, efficiency and speed [11]. More importantly, they did not rely on programmed temperature changes. Loop-mediated isothermal amplification (LAMP) [12,13], recombinase polymerase amplification (RPA) [14,15,16], hybridization chain reaction (HCR) [17] and recombinase-aided amplification (RAA) [18,19,20] were proposed and applied for bacterial detection. RAA received increasing attention since it could be completed under a constant low temperature (~39 °C), and was able to detect nucleic acids of *Salmonella* as low as 5 fg per reaction in 20 min [21]. RAA-based bacterial detection mainly includes the following two steps: DNA extraction and DNA detection. In general, DNA extraction was manually performed off-chip by well-trained technicians through first lysing bacterial cells to release DNA, then extracting DNA from lysate and finally purifying DNA to remove inhibitors (e.g., proteins and salts) [22,23,24,25]. For bacterial cell lysis, thermal lysis was a simple and effective method with no need for extra reagents and was extensively used in molecular biology analysis. Conventional thermal lysis was often carried out by first adding bacterial samples into a centrifugal tube and then heating them at 100 °C in a water or oil bath [26]. However, it generally required a long time to preheat the water or oil and incubate the bacterial samples. For nucleic acid extraction, magnetic silica beads (MSBs) were often used since they could well remove the background to avoid its negative effect on subsequent DNA detection [27], but they were easy to sediment due to their large size (micrometer level) and needed continuous vortex. Besides, DNA extraction and detection were basically separate, and this might lead to potential cross-contaminations. With the development of microfluidic technology, microfluidic systems provide a new solution to perform quantitative nucleic acid analysis in a shorter time by simply-trained technicians due to their low consumption and automatic control [28,29,30]. Some microfluidic chips were reported to combine with RAA for the detection of *Salmonella* or other foodborne pathogens, as summarized in Appendix A. These studies successfully achieved automated DNA amplification and detection on the chips; however, they still needed additional steps of bacterial lysis and DNA extraction off the chips. Hence, the integration of lysis, extraction, purification, amplification and detection is urgently needed to enable point-of-care testing of foodborne bacteria.

Here, we developed an integrated microfluidic chip from bacteria lysis to DNA extraction, amplification and detection (Figure 1A). As shown in Figure 1B, a spiral channel was first wound around an iron rod and heated with an eddy heater to continuous-flow lyse the bacterial cells, which released their nucleic acids, which were mixed with the MSBs using a 3D-printed fan-shaped mixer and separated from the lysate using a magnet. Then, the nucleic acids were eluted with the eluent to flow into the detection chamber. Finally, the extracted DNA was isothermal amplified and RAA detected to determine the bacterial amoun. The main novelty of this work is the microfluidic chip using non-contact eddy heating to continuous-flow lyse bacterial cells and the 3D-printed active mixing to continuous-flow mix the lysed DNA with MSBs. More importantly, the bacterial detection procedure, including lysis, mixing, separation, elution, amplification and detection, was integrated onto this single microfluidic chip.

## 2. Materials and Methods

### 2.1. Materials

*Salmonella typhimurium* (ATCC 14082) was used as target bacteria, *Escherichia coli*
*O157:H7* (ATCC 43888), *Listeria monocytogenes* (ATCC 13932) and *Staphylococcus aureus* (ATCC 25293) were used as non-target bacteria. For nucleic acid extraction and purification, the magnetic universal genomic DNA extraction kit was purchased from Tiangen Biotech (Beijing, China). For nucleic acid amplification, the RAA reaction kit was purchased from Amp-Future (Weifang, China). The EvaGreen fluorescent dye was purchased from Maokang Biotech (Shanghai, China).

Real-time quantitative PCR was performed using SsoFast™ EvaGreen Supermix on the CFX-384 PCR System (Bio-Rad, Hercules, CA, USA). The primer sets were designed based on the *Salmonella* invA gene’s conserved short fragments referring to the previous publication [21] and synthesized by Sangon Biotech (Shanghai, China). Bromocresol purple, sodium hydroxide and citric acid (Sinopharm, Wuhan, China) were used for evaluating the mixing performance. Luria-Bertani medium (Aoboxing Biotech, Beijing, China) was used for bacterial culture.

### 2.2. Development of Microfluidic Chip

The microfluidic chip was designed using Solidworks software, and the molds of the microfluidic chambers, channels and fan-shaped mixer were fabricated using an Object24 3D printer (Stratasys, Eden Prairie, MN). From Figure 1B, the chip mainly included a spiral lysis channel (volume: 100 μL) for lysing the bacterial cells and the following four chambers: (1) a cooling chamber (volume: 200 μL) for cooling the boiled lysate, (2) a mixing chamber (radius: 5 mm, height: 10 mm, volume: ~600 μL) for stirring the lysate and MSBs to form the MSB-DNA complexes, (3) a trapezoidal chamber (volume: ~50 μL) for separating and washing the complexes and eluting the DNA off the complexes, and (4) a detection chamber (200 μL centrifugal tube) for amplifying the purified DNA. The fan-shaped mixer was assembled inside this mixing chamber and rotated by the water flow. From Figure 1C, this microfluidic chip consisted of three layers (the fabrication process was shown in Appendix A). After each layer was fabricated by polydimethylsiloxane (PDMS), three layers were bonded together after surface plasma treatment. The bottom layer (length: 4.2 cm, width: 3.8 cm, height: 0.3 cm) contained a cylindrical hole (dia: 1.0 mm) to assemble a rotation axis for the fan-shaped mixer. The middle layer contained the following three components: (1) a spiral lysis channel (id: 0.8 mm, od: 1.2 mm, length: 20.0 cm) coiling around an iron rod (od: 5.0 mm, length: 1.5 cm), (2) an ellipsoidal cooling chamber (length: 7.5 mm, major axis: 4.0 mm, minor axis: 4.0 mm) and (3) a cylindrical chamber (dia: 1.0 cm, height: 5 mm) for housing the fan-shaped mixer at the center and two tangential inlets (length: 1.0 cm, width: 0.5 mm, height: 0.5 mm). The top layer contained a matching cylindrical hole to assemble the rotation axis, a trapezoid separation chamber (longer top-line: 7.0 mm, shorter bottom-line: 4.2 mm, height: 3.0 mm, thickness: 3.0 mm) to capture and wash the MSB-DNA complexes, and an outlet (dia: 5.6 mm) to connect with a centrifugal tube for RAA detection (Figure 1D).

### 2.3. Bacterial Cells Lysis and DNA Extraction

The bacterial cells were lysed by applying a voltage of 12 V on this self-developed eddy heater. The temperature was measured using both a temperature probe (NR81530, LiHuaDa, Shenzhen, China) and a smartphone infrared thermal imager (HT-102, XinTai, Dongguan, China). First, the microfluidic chip was placed on the eddy heater to preheat the iron rod for 30 s, making its temperature reach ~150 °C. Then, 1 mL bacterial sample was centrifuged at 10,000 rpm for 5 min, mixed with 100 µL DNA extraction buffer and injected into the microfluidic chip at different flow rates using a syringe pump (11Elite, Harvard Apparatus, FL, USA). Finally, the lysate was mixed with 15 μL MSBs to form the MSB-DNA complexes, followed by magnetic separation for 2 min, cleaning with washing buffer for 4 times and elution with 50 μL eluent to obtain the purified DNA, which was determined using quantitative PCR to evaluate the performance of DNA lysis. The sequences for the primer set were listed in Appendix A and PCR reaction was performed as follows: 95 °C for 30 s, 40 cycles of 95 °C for 5 s and 60 °C for 30 s [31]. All the samples were tested in triplicates using the same protocol.

### 2.4. DNA Separation and RAA Detection

After the MSB-DNA complexes were formed in the mixer, they were first captured in the trapezoid separation chamber by the magnet. Then, washing buffer was injected to remove the residual, followed by injecting the air at 2 mL/min for 3 min to accelerate the ethanol volatilization and 50 μL eluent (deionized water) to elute the DNA. Finally, RAA reaction was performed at 39 °C in a total volume of 50 μL comprising 29.4 μL RAA enzymes buffer A, 11.6 μL target DNA templates, 2 μL 10 μM forward primers, 2 μL 10 μM reverse primers, 2.5 μL EvaGreen fluorescent dye (20×) and 2.5 μL 280 mM magnesium acetate. The fluorescent signals were collected every 1 min (λ_ex_: 445 nm; λ_em_: 526 nm) for 25 min in a microplate fluorescence reader (infinite M NANO, Tecan, Swiss), and the fluorescence spectrum of the EvaGreen fluorescent dye from the fluorescence reader was shown in Appendix A. The threshold time (*T_t_*) was determined as the time for the fluorescence intensity to exceed the sum of the average intensity of first 8 min plus 3× standard deviations.

### 2.5. Spiked Sample Preparation and Detection

The preparation of spiked pork samples was referring to China’s food safety national standards GB 4789.4-2016. Briefly, 25 g pork meats were first sliced and diluted with 225 mL sterile phosphate buffer, then homogenized with a stomacher and stood for 15 min to obtain the supernatant, and finally mixed with different concentrations (1.1 × 10^2^–10^5^ CFU/mL) of bacteria cells to prepare the spiked pork samples, which were detected using both this microfluidic chip and the gold standard culture plating method. For bacterial detection using this microfluidic chip, after each prepared sample was first injected into the lysis channel at optimized flow rate (0.5 mL/min) to release the nucleic acids, the lysate was then continuous-flow mixed with the MSBs at optimized fluid rate (10 mL/min) to extract the nucleic acids, which were finally amplified and detection by RAA.

## 3. Results and Discussion

### 3.1. Effect of Eddy Heating for Bacterial Cells Lysis

Bacterial cell lysis is the precondition to nucleic acid extraction. In this study, heating lysis was employed for continuous-flow lysis the bacterial cells using an eddy heater. This heater was simply developed based on Faraday’s law of electromagnetic induction by using an iron rod and an alternating electromagnetic field generator (ZVS, QICHUANG, Qingdao, China). When the iron rod was exposed to the alternating electromagnetic field, a large eddy current was generated inside the iron rod due to the small resistance of iron, and thus the iron rod was rapidly heated up to a high temperature, which was estimated using the smartphone infrared thermal imager. The temperature change of the iron rod at a different time and different voltage, as shown in Appendix A, which the temperature could reach up to ~150 °C in 30 s when the voltage of 12 V was applied. As shown in Figure 2A, the Teflon tubing was tightly wound around the iron rod to form the spiral channel, and deionized water was injected into the channel to accurately measure its actual temperature in the channel as ~100 °C using the thermocouple probe (shown in Appendix A). Figure 2B showed the simulation results using the ANSYS software (Canonsburg, PA, USA), which were consistent with the measured temperature. Besides, an ellipsoidal cooling chamber was used to collect the lysate and allow it to standing for 1 min to decrease the temperature to ~75 °C, which was also confirmed using this thermocouple probe. Therefore, when the mixture of bacterial sample and lysis buffer was injected to continuously flow through this spiral channel, the mixture was kept at a high temperature for a certain time, which was determined by the length of this channel at a constant flow rate, allowing the sufficient lysis of bacterial cells. Thus, different flow rates from 0.1 to 3.0 mL/min were used to lyse the *Salmonella* cells at 1.1 × 10^6^ CFU/mL by this eddy heater in the presence of lysis buffer, followed by extraction of nucleic acids through a manual MSB-based method and detection of nucleic acids through real-time PCR. As shown in Figure 2C, the Ct value increased from 22.9 to 23.3, when the flow rate decreased from 0.5 to 1 mL/min. This was mainly because the higher flow rate of the bacterial sample shortened the heating time for destroying the bacterial cells and releasing their DNA. A further decrease in the flow rate to 0.1 mL/min did not lead to an obvious increase in the Ct value. Thus, the flow rate was optimized at 0.5 mL/min. After the temperature was generated up to 150 °C and the aqueous bacterial sample was injected at 0.5 mL/min into the tubing, the sample was observed to be first evaporated into a gas at the fourth round and then returned back to liquid at the cooling chamber (Appendix A).

To ensure the sufficient lysis of bacterial cells, *Salmonella* cells at 1.1 × 10^7^ CFU/mL were heated and lysed at different times to release the nucleic acids, which were determined using real-time qPCR. From Figure 2D, when the time was changed from 6 to 12 s, the Ct value significantly decreased from 19.3 to 17.8 (*p* < 0.05), indicating more cells were lysed and thus more nucleic acids were obtained. Besides, a further increase in the time to 24 s did not lead to an obvious decrease in the Ct value. This showed that the time of 12 s was sufficient for heating the bacterial cells. Therefore, the length of this spiral channel was calculated as the product of the optimal time and the flow rate to be 20 cm. Table 1 showed the comparison with the existing chemical or other lysis methods. This heating lysis was significantly faster (~12 s) and could continuous-flow lyse the bacterial samples. This might be because the bacterial cells in the fluidic channel suffered from more homogeneous heating and higher temperatures by this eddy heater, resulting in an easier rupture of bacterial cells.

To further demonstrate the advantage of this continuous-flow heating lysis method over the conventional static one, the bacterial cells at 1.1 × 10^4^–1.1 × 10^7^ CFU/mL were in parallel lysed using this heating lysis method at 0.5 mL/min and the conventional one in centrifugal tubes, and real-time qPCR was used to amplify the lysate for determination of the released DNA. The same concentrations of bacterial samples were heated at 100 °C for 10 min and used as positive controls, and deionized water was used as a negative control. From Figure 2E, the Ct values for this continuous-flow lysis were slightly lower than those for the conventional lysis, indicating that more nucleic acids were released due to more lysed bacterial cells by this continuous-flow heating lysis. Especially, for the high concentration (1.1 × 10^7^ CFU/mL) of bacterial cells, the mean Ct value of triple tests for this continuous-flow lysis was significantly lower (*p* < 0.05), and the lower Ct values indicated that more nucleic acids were lysed from the bacterial cells. Statistical analysis was performed using SPSS (IBM SPSS 22, New York, NY, USA).

### 3.2. Performance of 3D Fan-Shaped Mixer

The mixing of the MSBs with the lysate played a crucial role in nucleic acid extraction. Since the large-size of MSBs (~1 μm) were easy to settle down due to their gravity, resulting in the reduced chances to adsorb the DNA in the lysate, the active fan-shaped mixer was 3D-printed and assembled in the center of this mixing chamber (Figure 3A). When the lysate and MSBs were injected into the mixing chamber to hit the blades of this mixer, the mixer automatically rotated on the axis, resulting in effective mixing of the lysate and MSBs in the continuous-flow condition. Besides, to avoid the sedimentation of MSBs, the lysate and MSBs were designed to enter this mixing chamber from its bottom and their mixture was designed to leave this chamber from its top. Appendix A was recorded to show the sufficient mixing of MSBs with the red ink. To verify this, the dynamic flow simulation was conducted using the ANSYS software based on the Reynolds Average Navier-Stokes equations. Figure 3B showed a clear vortex flow, which might lead to sufficient mixing in the whole chamber.

The rotation of this fan-shaped mixer is the precondition for sufficient mixing of the continuous-flow solutions and has to overcome the sliding friction and fluidic resistance. Thus, different flow rates were applied to check the minimum one for rotating the mixer. As shown in Appendix A, when the flow rate was gradually increased from 2 to 6 mL/min, the mixer remained static because the driving force resulting from the hit on the blade was still smaller than the slide friction and fluidic resistance. When the flow rate was increased to 6 mL/min, the mixer started to rotate slowly and became faster at higher flow rates. Thus, the flow rate had to exceed 6 mL/min to rotate this mixer.

To test the mixing efficiency of this mixer, bromocresol purple with 0.3 M sodium hydroxide (NaOH) and 0.1 M citric acid was injected from two inlets into this mixing chamber at different flow rates, respectively. Besides, the mixing chamber without this mixer was used for comparison. The color of bromocresol purple was observed to shift from purple (pH 5.2) to yellow (pH 6.8), which was recorded and analyzed by ImageJ software. The mixing rate (MR) is defined as the ratio of standard deviation to mean value, i.e.,
$$\mathrm{MR}=\left(1-\frac{\sqrt{\frac{1}{N}\sum_{i=1}^{N}{\left(I_{i}-\bar{I}\right)}^{2}}}{\bar{I}}\right)\times 100\%$$
where, I_i_ is the pixel intensity of point i, Ī is the mean intensity of all the points, and N is the total number of points. Figure 3C showed that the mixing rate increased with the flow rate because a faster flow rate accelerated the rotation of this mixer and thus improved the mixing effect. Besides, the rates for the mixing chamber with this mixer were obviously higher than those without this mixer, indicating this active mixer enhanced the mixing efficiency. Besides, the mixing rate at 10 mL/min was more than 95% and basically remained the same level for higher flow rates. Thus, the flow rate was optimized at 10 mL/min.

The time for mixing the released nucleic acids with the MSBs is also vital for DNA extraction. Thus, the nucleic acids of *Salmonella typhimurium* at 100 ng/μL, which were extracted using enzyme lysis and MSBs purification and determined using a Nanodrop 2000 spectrophotometer (Thermo Fisher, Waltham, MA, USA), were simultaneously injected with the MSBs into the mixing chamber, and a peristaltic pump (Kamoer Fluid Technology, Shanghai, China) was used with a three-way connector to recycle the continuous-flow mixing for a different time, followed by magnetic separation and DNA elution to obtain the extracted DNA, which was determined using the Nanodrop spectrophotometer to calculate the capture efficiency, i.e., the ratio of the extracted DNA to the original one. From Figure 3D, the capture efficiency increased with the mixing time from 0 to 7 min and reach ~90% at a mixing time of 7 min or longer. Therefore, the mixing time was optimized to be 7 min.

### 3.3. Performance of Microfluidic Chip

For quantitative detection of unknown concentrations of *Salmonella typhimurium* in a sample, different concentrations of viable *Salmonella typhimurium* at 1.1 × 10^2^–1.1 × 10^5^ CFU/ mL were detected under optimal conditions using this microfluidic chip and the standard culture plating method to establish its mathematical model. All the tests were conducted in triplicate. From Figure 4A, the time threshold decreased from 7.1 to 6.0 min as the bacterial concentration increased from 1.1 × 10^2^ to 1.1 ×10^5^ CFU/ mL. A linear relation between time threshold (T_t_) and bacterial concentration (C) was observed and can be expressed as T_t_ = −1.37×lg(C) + 16.58 (R^2^ = 0.9954). The lower detection limit was calculated as 89 CFU/mL, referring to 3× SNR.

To evaluate its specificity, other foodborne pathogens (*Listeria monocytogenes,*
*Escherichia coli*
*O157:H7* and *Staphylococcus aureus*) at 1.1 × 10^5^ CFU/mL were tested and compared to *Salmonella typhimurium* at 1.1 × 10^4^ CFU/mL. Besides, a negative control (deionized water) was also detected. The original amplification curves were obtained from the fluorescence reader and shown in Appendix A, and it was clearly observed that the fluorescence intensities of the background were different, which might lead to a difficult analysis of the fluorescent signals. To minimize the impact of the background, the fluorescence intensity of each point (F) in the amplification curve was normalized by the following equation: F = (F_i_ − F_0_)/(F_m_ − F_0_), where F_0_ is the initial intensity at the starting time (t = 1 min); F_m_ is the maximum intensity at the ending time (t = 25 min); F_i_ is the fluorescent intensity measured using the reader. From Figure 4B, only *Salmonella typhimurium* showed a fluorescence signal in 10 min, while the other bacteria and negative control did not have significant signals, indicating that it had good specificity.

To further evaluate its applicability, the spiked pork samples with known bacterial concentrations were detected. The recovery (R) was calculated as the ratio of the detected concentration (C_d_) to the added one (C_a_), i.e., R= C_d_/C_a_ × 100%. As shown in Table 2, the recovery for different concentrations ranged from 84.3% to 103.2% with an average recovery of 94.1%, indicating it could be used for practical detection of *Salmonella typhimurium* in real pork samples.

The time for the whole bacterial detection procedure was 40 min, including 1 min for lysis, 1 min for cooling, 7 min for mixing, 5 min for washing and 25 min for detection. The excellent performance of this microfluidic chip could be attributed to the following aspects: (1) effective bacterial cell lysis using this continuous-flow eddy heating; (2) sufficient mixing using this fan-shape active mixer; (3) accurate DNA detection using this RAA method. More importantly, the whole bacterial detection procedure, including lysis, mixing, washing, capture, separation and detection, was integrated onto one single chip and completed in a short time.

## 4. Conclusions

In this work, we successfully explored a microfluidic chip and combined it with RAA for quantitatively detecting *Salmonella*
*typhimurium*. This eddy heating was demonstrated with the ability to effectively lyse the flow-through bacterial cells. The 3D-printed fan-shaped active mixer was verified to sufficiently mix the lysate and MSBs. The bacterial lysis, DNA purification and RAA detection were integrated into one single chip to achieve sample-in-result-out detection of *Salmonella* as low as 89 CFU/mL. This microfluidic chip was featured with low cost (<$3/chip and ~$150/peripheral), short time (40 min from sample-in to result-out) and minimal cross-contamination (closed microfluidic chip and non-contact heating). It might be further improved by miniaturizing the device and chip for point-of-care testing of foodborne pathogens.

## Figures and Tables

**Figure 1 biosensors-12-00726-f001:**
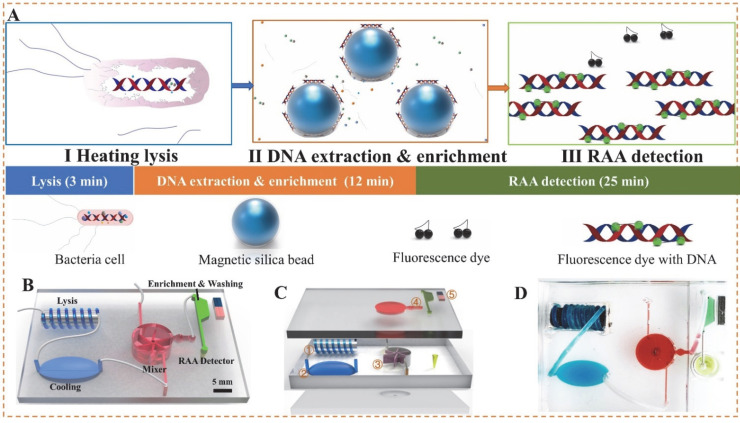
The schematic of this microfluidic chip for *Salmonella typhimurium* detection. (**A**) The procedure of bacterial detection. (**B**) The structure of the microfluidic chip. (**C**) The components of the microfluidic chip: ① the spiral lysis channel, ② the ellipsoidal cooling chamber, ③ the fan-shaped mixer, ④ the trapezoid separation chamber, ⑤ the magnet. (**D**) The photo of the microfluidic chip.

**Figure 2 biosensors-12-00726-f002:**
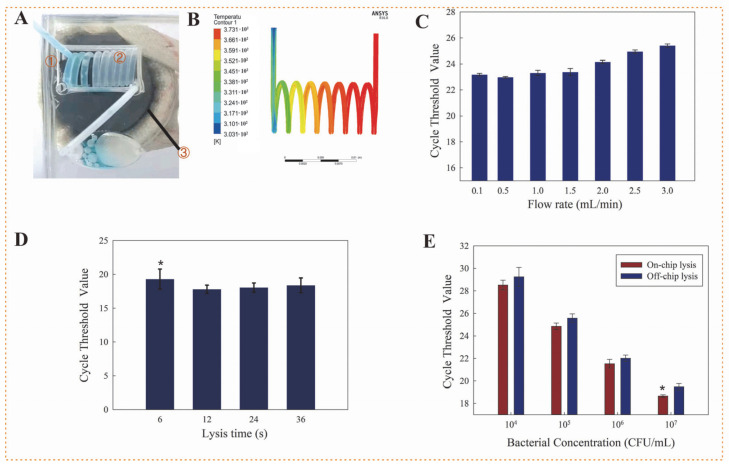
The eddy heating for bacterial cells lysis. (**A**) The photo of the eddy heater: ① the iron rod, ② the spiral channel, ③ and the alternating electromagnetic field generator. (**B**) The simulation on the temperature of deionized water in the spiral channel (unit: K). (**C**) The Ct values of *Salmonella* cells at 1.1 × 10^6^ CFU/mL for different flow rates (N = 3). (**D**) The Ct values of *Salmonella* cells at 1.1 × 10^7^ CFU/mL for different lysis time (N = 3). (**E**) The Ct values for different concentrations of *Salmonella* cells at the flow rate of 0.5 mL/min (N = 3). * indicated significance difference, *p* < 0.05.

**Figure 3 biosensors-12-00726-f003:**
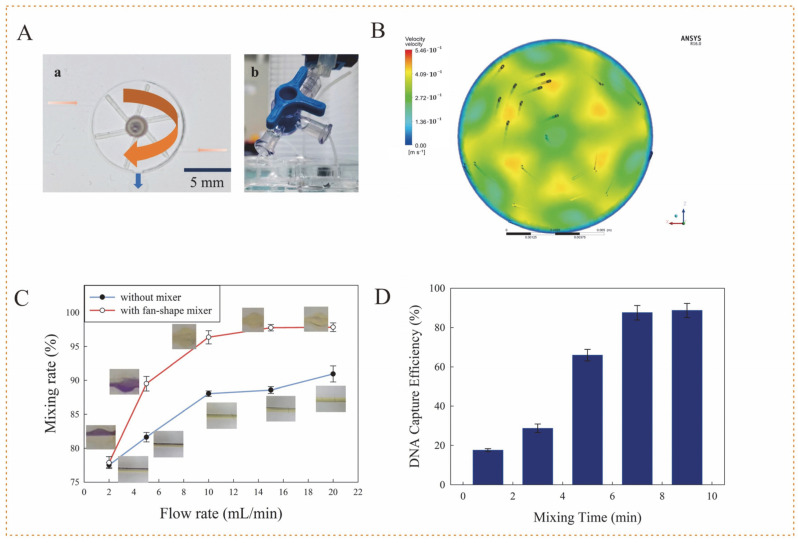
The mixing performance of the 3D-shaped mixer. (**A**) The structure of this 3D-printed fan-shaped mixer: (a) The photo of this mixer; (b) The three-way connector. (**B**) The simulation results of the fan-shaped mixer. (**C**) The mixing rate of the chambers with and without the mixer at different flow rates (N = 3). (**D**) The DNA capture efficiency for different mixing time at 10 mL/min (N = 3).

**Figure 4 biosensors-12-00726-f004:**
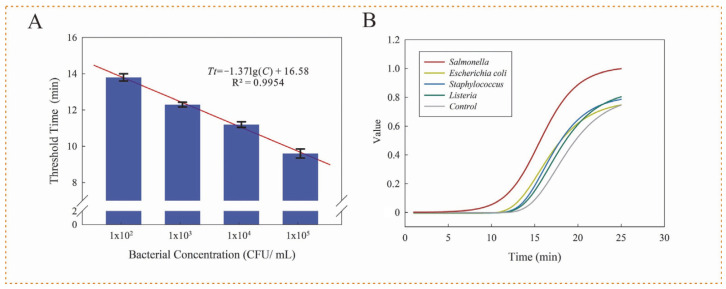
Performance of this microfluidic chip. (**A**) The linear relationship between the threshold time and the bacterial concentration (N = 3). (**B**) The specificity of this microfluidic chip.

**Table 1 biosensors-12-00726-t001:** Comparison of this lysis method with recently foodborne pathogens lysis method.

Methods	Targets	Lysis Time	Amount/Volume	References
Photothermal lysis	*Escherichia coli* O157:H7	6 min	5.0 × 10^5^ CFU/50 μL	[32]
Chemical lysis	*Salmonella typhimurium*	10 min	5.0 × 10^2^ CFU/72 μL	[33]
Mechanical lysis	*Enterococcus faecalis*	6 min	1.0 × 10^6^ CFU/100μL	[34]
Electrical lysis	*Salmonella enterica*	40 s	1.0 × 10^6^ CFU/100 μL	[35]
This method	*Salmonella typhimurium*	12 s	1.1 × 10^7^ CFU/100 μL	

**Table 2 biosensors-12-00726-t002:** The recovery of the spiked *Salmonella typhimurium*.

Added Concentration (CFU/mL)	Detected Concentration (CFU/mL)	Recovery Rate (%)
110	104.3 ± 21.6	94.8% ± 19.6%
1100	1135.0 ± 189.0	103.2% ± 17.2%
11,000	9277.2 ± 1544.7	84.3% ± 14.0%

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
