# Peer review of "A Salmonella Microfluidic Chip Combining Non-Contact Eddy Heater and 3D Fan-Shaped Mixer with Recombinase Aided Amplification"

_biosensors, 2022, doi:10.3390/bios12090726_

Round 1

Reviewer 1 Report

  • Review Summary

    Title: A Salmonella Microfluidic Chip Combining Non-Contact Eddy Heater and 3D Fan-Shaped Mixer with Recombinase Aided Amplification

    Manuscript ID: biosensors-1851670

    Article Summary: The authors developed a microfluidic chip for integrated lysis and detection of salmonella by combining thermal lysis using a non-contact eddy heater, magnetic bead separation of bacterial DNA aided by a fan-shaped heater, and nucleic acid detection using isothermal recombinase aided amplification. The authors claim they can quantitatively detect Salmonella from 1.1 x 10^2 to 1.1 x 10^5 CFU/mL in 40 min with a detection limit of 89 CFU/mL.

    Reviewer Recommendation: The manuscript may be publishable after minor revisions. The authors combine multiple components into an elegant demonstration. Additional context for this work would help to better convey the significance of this work to readers and clarify how it compares with previous demonstrations in the field. In addition, additional technical details and analysis of this paper would improve clarity and readability as described in detail below.

  • Major Comments

    • Provide more examples of microfluidic automation and integration of DNA extraction and amplification beyond systems focused on RAA.
      • Lines 70 - 71 only talk about RAA but what about other microfluidic solutions that have been developed for Salmonella or other bacteria detection. How does this work compare with the state of the art in microfluidic bacteria detection?
    • The authors talk about how assay steps were integrated on chip but in lines 136 - 137 describe centrifuging the bacterial sample prior to addition to the microfluidic chip.
      • Is centrifugation necessary prior to on-chip lysis and detection?
      • How do the authors plan to integrate this process into the microchip?
    • The comparison in table 1 is a little misleading
      • I think it would be more accurate to compare the lysis time required for similar amounts of bacteria.
      • For example, the authors claim to lyse 1.1 x 10^7 CFU in 12s, but how much time would be required to lyse 5.0 x 10^2 CFU. I would assume longer than 12s.
        • Does the time required increase linearly with bacteria concentration?
        • Based on the Ct values in Figure 2E, I wonder how a 12s lysis time would perform with only 10^2 CFU of bacteria.
    • Are there supposed to be supplementary videos?
      • I only saw the supplementary PDF but not any videos on the peer review page.
    • How was flow rate controlled and adjusted for differing requirements between lysis and mixing regions?
      • Lines 193 - 197 indicate that lysis requires 0.5 mL/min flow rate and that Ct values increase when flow rate is 1 mL/min
      • Lines 251 - 252 indicate that flow rate must be > 6 mL/min to rotate the mixer. How was flow controlled and adjusted during different stages?
        • Was this automated?
        • How will this be integrated into a portable and inexpensive format suitable for point of care use?
    • What is the clinically recommended target concentration for detecting Salmonella in clinical samples?
      • Will there be scenarios where ≤10 CFU/mL need to be detected?
        • How do the authors plan to address such scenarios?
      • The authors do not discuss the impact of sample volume on limit of detection?
        • How does the choice of sample volume affect the lowest detection limit achievable with the device?
        • For example if 100 µL of sample is screened, what is the theoretical lowest limit of detection achievable with the instrument? How does the performance of their device compare with this?
  • Minor Comments

    • Figure 1B - add a scale bar to the picture of the microfluidic chip
    • Figure 3B is very unclear - the legend and text are very grainy and unreadable
      • please include a higher resolution and larger image
    • Figure 3C - the inset images are very small and hard to see
      • please include larger and higher resolution insets

Author Response

We very much appreciate you for the review on our manuscript with important comments and constructive suggestions for us to greatly improve this manuscript. Your comments have been carefully considered point by point and the revisions have been made in the revised manuscript according to your suggestions. The detailed point-by-point responses to your comments have been attached. Thank you very much.

Reviewer 2 Report

This paper reports a microfluidic chip technology for rapid and sensitive detection of Salmonella Typhimurium. The major novelty of this paper is the eddy-heating-based cell lysis. Overall, the structure of the paper is good and the results are helpful for developing a simple, low-cost, fast and specific bacterial 28 detection technique for ensuring food safety. I suggest accepting this paper if the authors can make revisions according to the following comments:

1.       The structure and the detailed fabrication procedures for the chip shown in Figure 1B should be clearly presented. Currently, it is very hard to know how the components of the chip were fabricated.

2.       The time for cell lysis: This is the most important advantages of this method. However, no explanations about why this can be achieved in only 12s.

3.       No label for the Y-axis of the Figure 4B.

4.       Section 3.3:no descriptions how the pork samples were prepared and measured.

5.       In this paper, only the structure of the chip was described. How the fluorescence signals were measured? Are they measured with a commercial instrument? Or a customer-made device?

Author Response

(The authors gave the same response as above.)

Round 2

Author Response

We very much appreciate you for the review on our manuscript again with important comments and constructive suggestions. Your comments have been carefully considered point by point and the revisions have been made in the revised manuscript according to your suggestions. The detailed point-by-point responses to your comments have been attached. Thank you very much.

Round 3

Reviewer 2 Report

The author addressed my concern and made revisions. It can be accepted for publication.